# Renal Endothelial Single-Cell Transcriptomics Reveals Spatiotemporal Regulation and Divergent Roles of Differential Gene Transcription and Alternative Splicing in Murine Diabetic Nephropathy

**DOI:** 10.3390/ijms25084320

**Published:** 2024-04-13

**Authors:** Alex-Xianghua Zhou, Marie Jeansson, Liqun He, Leif Wigge, Pernilla Tonelius, Ramesh Tati, Linda Cederblad, Lars Muhl, Martin Uhrbom, Jianping Liu, Anna Björnson Granqvist, Lilach O. Lerman, Christer Betsholtz, Pernille B. L. Hansen

**Affiliations:** 1Research and Early Development, Cardiovascular, Renal and Metabolism, BioPharmaceuticals R&D, AstraZeneca, Gothenburg, 43162 Mölndal, Sweden; alex.zhou1@astrazeneca.com (A.-X.Z.); pernilla.tonelius@astrazeneca.com (P.T.); martin.uhrbom@astrazeneca.com (M.U.);; 2Department of Medicine Huddinge, Karolinska Institutet, 141 52 Huddinge, Sweden; marie.jeansson@ki.se (M.J.); jianping.liu@ki.se (J.L.);; 3Department of Immunology, Genetics and Pathology, Uppsala University, 753 10 Uppsala, Sweden; 4Data Sciences and Quantitative Biology, Discovery Sciences, BioPharmaceuticals R&D, AstraZeneca, Gothenburg, 43162 Mölndal, Sweden; 5Division of Nephrology and Hypertension, Mayo Clinic, Rochester, MN 55902, USA; lerman.lilach@mayo.edu

**Keywords:** diabetic nephropathy, transcriptomics, endothelium

## Abstract

Endothelial cell (EC) injury is a crucial contributor to the progression of diabetic kidney disease (DKD), but the specific EC populations and mechanisms involved remain elusive. Kidney ECs (*n* = 5464) were collected at three timepoints from diabetic BTBR*ob/ob* mice and non-diabetic littermates. Their heterogeneity, transcriptional changes, and alternative splicing during DKD progression were mapped using SmartSeq2 single-cell RNA sequencing (scRNAseq) and elucidated through pathway, network, and gene ontology enrichment analyses. We identified 13 distinct transcriptional EC phenotypes corresponding to different kidney vessel subtypes, confirmed through in situ hybridization and immunofluorescence. EC subtypes along nephrons displayed extensive zonation related to their functions. Differential gene expression analyses in peritubular and glomerular ECs in DKD underlined the regulation of DKD-relevant pathways including EIF2 signaling, oxidative phosphorylation, and IGF1 signaling. Importantly, this revealed the differential alteration of these pathways between the two EC subtypes and changes during disease progression. Furthermore, glomerular and peritubular ECs also displayed aberrant and dynamic alterations in alternative splicing (AS), which is strongly associated with DNA repair. Strikingly, genes displaying differential transcription or alternative splicing participate in divergent biological processes. Our study reveals the spatiotemporal regulation of gene transcription and AS linked to DKD progression, providing insight into pathomechanisms and clues to novel therapeutic targets for DKD treatment.

## 1. Introduction

DKD is an increasing healthcare challenge worldwide imposing a large, unmet medical need. About 40% of DKD is likely progressed from the heterogenous diabetic patient population [1]. The lack of longitudinal data from patients and animal models limits the understanding of the disease-driving mechanisms and hinders the development of novel therapies. Specifically, EC dysfunction and capillary rarefaction are commonly observed in DKD, correlating with impaired kidney function and predicting future development of end-stage renal disease [2].

Little is known about changes in EC gene expression accompanying DKD progression, and how such changes over time correlate with specific EC subtypes and spatial location. The molecular heterogeneity of ECs has been investigated recently via scRNAseq [3,4,5,6]. These studies characterized murine renal ECs at developmental and physiological stages [3] or during adaptation to dehydration [4], whereas the underlying mechanisms driving endothelial damage and DKD progression remain unresolved. Limited information is available on ECs from the whole kidneys or glomerular single-cell transcriptomes in mouse DKD [7,8,9,10]. Notably, none of these studies have distinguished transcriptional and post-transcriptional processes. In humans, approximately 95% of multiexon genes undergo AS [11], which adds another layer of gene expression regulation. Importantly, a recent scRNAseq study [12] demonstrated that splice isoform switching is critical in the mesenchymal–epithelial transition in kidney development. This underscored the feasibility and importance of exploring differential AS in specific cell types during DKD progression.

In the current study, we utilized diabetic BTBR*ob*/*ob* mice, a model of obesity-induced type-2 diabetes that develops early pathological features of human DKD [13,14]. The leptin-deficient BTBR*ob*/*ob* mice feature hyperglycemia, albuminuria, obesity, and hyperlipidemia, but not hypertension [14,15,16]. We applied SmartSeq2 full-length scRNAseq to ECs at three timepoints to investigate the specific transcriptional response and aberrant AS in different vascular compartments in DKD progression.

## 2. Results

### 2.1. Molecular Profiling and EC Populations

To characterize the molecular heterogeneity of renal ECs and changes during DKD progression, we analyzed kidney ECs from 6/11/20-week-old diabetic BTBR*ob*/*ob* and non-diabetic BTBRLean mice. The experimental design is summarized in Figure 1A. Renal ECs were isolated using Pecam1 staining and FACS, which also demonstrated lower numbers of ECs in 11- and 20-week-old BTBR*ob*/*ob* mice (Figure 1B). BTBR*ob*/*ob* mice displayed significantly increased blood glucose and albuminuria levels at 11 weeks of age (Figure 1C,D). The resulting dataset contained transcriptomes of 5464 cells covering different genotypes, EC subtypes, and timepoints (Appendix A).

A uniform manifold approximation and projection (UMAP) of ECs from BTBRLean mice identified 13 EC clusters (Figure 1E,F), expressing EC markers (Figure 1G) [17,18] but no other renal cell markers (see Methods). For EC subtype annotation (Figure 1H), we compared cluster-specific gene patterns with published renal EC scRNAseq data and protein and mRNA localization in situ [3,4,5]. This annotated the 13 clusters as arteries/afferent arterioles (AA), glomerular (GEC), efferent arteriole (EA), ascending vasa recta (AVR), descending vasa recta (DVR), peritubular capillaries (PCEC), venous (VEC), proliferating (cycling), tip cells, and lymphatic (LVEC). One unknown cluster remained. As an example, *Calca* was localized to efferent arteriole, which was confirmed through RNA-ISH in C57BL6/J mice (Figure 1I).

### 2.2. Results from Renal EC Heterogeneity, Annotations, and Validation

As most EC populations did not manifest unique singular markers, we performed RNA-ISH and immunofluorescence (IF) in C57BL6/J mice against a combination of markers to determine EC identity. AA was characterized by high expressions of *Fbln5*, *Gja5*, *Cxcl12*, and *Cldn5* and a low expression of *Plvap* [3,4]. RNA-ISH for both *Cldn5* and Cldn5-GFP reported expressions showing strong signals in AA (Appendix A). RNA-ISH also indicated *Cldn5* expression in podocytes, which confirms previous findings [5,19]. However, Cldn5-GFP did not display a detectable GFP signal in podocytes (Appendix A), suggesting that relevant cis-acting promoter sequences for podocyte expression is missing in this transgenic construct. We identified *Ace* as a novel marker for AA, as confirmed through RNA-ISH (Appendix A). GEC showed a weak *Ace* expression, as previously reported [20].

EA differed substantially from AA, and while they express *Fbln5*, they have a very low *Cldn5* expression and no *Ace* expression. *Calca* appears to be specific to EA (Figure 1I), as previously reported [4]. In addition, *Tspan8* and *Npnt* were highly expressed in EA/DVR; however, some datasets report non-EC kidney expressions of these markers [21].

DVR tightly control perfusion to the outer and inner medulla and function both as exchange (capillary) and resistance (arteriolar) vessels [22]. DVR shared certain markers with AA, including *Gja5* and *Cldn5*, but also expressed the specific marker *Slc14a1* (Appendix A). Among ECs, DVR showed a high and specific expression of *Aqp1*, which was otherwise expressed primarily in epithelial cells (Appendix A).

The GEC/PCEC/AVR endothelium had fenestrations [23,24], but only fenestrations in PCEC/AVR had PLVAP+ diaphragms (Appendix A) [25], while healthy GEC lacked diaphragms [26]. Immunohistochemistry staining confirmed Plvap expression in PCEC/AVR, while *Cldn5* was absent (Appendix A). We identified *Il33* as a novel transcript in PCEC (Appendix A), with expression also in VEC/AVR, but not in GEC. *Car8* was expressed in PCEC/AVR/VEC, and colocalized with *Plvap* in AVR, whereas DVR (*Cldn5*+) were *Plvap*-negative (Appendix A). The *Car8* expression in VEC has been reported previously [4]. *Igf1* was expressed in AVR/PCEC and colocalized with *Plvap*. DVR (*Pecam1*+, *Plvap*-) were negative for *Igf1* (Appendix A). GEC were annotated by previously reported markers, including high expressions of *Gata5*, *Ehd3*, *Tbx3, Plat*, *Lpl*, *Kcnj5*, and *Adamtsl5* and low expressions of *Cldn5* and *Plvap* [3,4,5]. RNA-ISH for *Gata5* showed strong and specific staining in GEC (Appendix A). All data are presented in a searchable database with data output in bar plots (see the examples for *Cldn5* and *Plvap* in Appendix A).

Endothelial cell types in human kidney were conserved in the mouse kidney [6]. The gene expression profiles of GECs and PCECs in our dataset correlated well with recently described human GECs and PCECs [27] in spite of certain species-specific genes (Appendix A), suggesting cross-species translatability of our mouse data. Furthermore, 10 of 53 genes associated with human DKD using GWAS [28] could be mapped to the EC expression in our dataset (Appendix A). Of these, *Igfbp5* and *Tspan9* showed abundant EC expression: *Igfbp5* was enriched in GEC/PCEC/AVR, while *Tspan9* was expressed in all clusters (Appendix A). A comparison with the human Nephrocell whole-kidney scRNAseq database [29] confirmed the enrichment of *IGFBP5* in GEC/PCEC, in contrast to a ubiquitous expression of *TSPAN9* (Appendix A).

We noticed two clusters each for GEC and PCEC. One of each had relatively higher levels of immediate early gene transcripts (IEG) (GEC^IEG-high^ and PCEC^IEG-high^, respectively). IEGs are induced within minutes in response to cell-extrinsic and cell-intrinsic signals and do not require de novo protein synthesis for their enhanced expression [30,31]. RNA in situ hybridization (RNA-ISH) for IEGs including *Fos*, *JunB*, and *Atf3*, did not show expression patterns specific to any anatomical location, such as cortical versus juxtamedullary glomeruli (Appendix A). A high IEG expression has been reported in a proportion of vascular cells in scRNAseq data and suggested to result from stress and hemodynamic changes during tissue dissociation and cell isolation [32,33].

### 2.3. Functional Zonation of the Healthy Renal Vasculature

In BTBRLean mice, we identified differentially expressed genes (DEGs) in each EC cluster relative to the remaining clusters (Appendix A). An Ingenuity Pathway Analysis (IPA) of the DEGs displayed a complex zonation of pathway activation, related to the immune responses, vascular tone, coagulation, and growth factor signaling (Figure 2A, Appendix A).

ECs communicate with immune cells through chemokines, cytokines, and adhesion molecules. Chemokines showed enriched expressions in AA/GEC, including *Cxcl1*, *Cxcl12*/*Sdf1*, *Cxcl16*, and *Cx3cl1* (Figure 2B). Interleukins *Il16* and *Il33* displayed higher expressions in DVR and VEC, respectively. Adhesion molecules *Selp* and *Sele* were expressed in small portions of AA and AVR, respectively, while *Icam1* and *Vcam1* were enriched in VEC. Furthermore, MHC class-II molecules and *Cd74* were enriched in PCEC, while *Lbp*, in VEC. The anatomically defined patterns of immune genes within the EC compartments may contribute to the nephron’s immune zonation [34].

EC Weibel–Palade bodies regulate coagulation, and their components *Vwf*, *Selp*, *Edn1*, and *CD63* were mainly expressed in AA/DVR (Figure 2C). GEC and AA expressed *Plat* and its inhibitor *Serpin1*, respectively. Coagulation Factor *F8* was enriched in GEC. EC-released prostaglandins inhibit platelet aggregation: the prostaglandin synthase *Ptgs1* and transporter *Slco2a1* were expressed in VEC and AA, respectively. These findings suggest a differential initiation of coagulation and platelet activation along nephrons.

Regulators for vasodilation and constriction were enriched in AA, suggesting an EC zonation of vascular tone regulation (Figure 2D). Ca^2+^-calmodulin-dependent *Nos3* and Ca^2+^-activated *Kcnn3* and *Kcnn4* trigger hyperpolarization [35], spreading via *Gja4*/*Gja5* to elicit vasodilation in VSMCs [36,37]. *Gja4*/*5* also affect renin secretion in granular cells [38], and together with *Ace*, may coordinate the conversion of angiotensinogen. EC Ca^2+^ may trigger the release of *Edn1* [39] in AA/EA/DVR. Furthermore, regulators for intracellular Ca^2+^ homeostasis were enriched in AA/EA, including *Atp2a3*, *Trpv4*, *F2rl1*, and *Gper1* [40], and for extracellular Ca^2+^ homeostasis in AA/GEC/EA, these included *Pthlh, Calca*, *Sost* [41], and *Mgp*. Our study suggests that glomerulus-linked (AA/GEC/EA) and DVR zonation fine-tunes renal vascular tone.

The IGF system displays a comprehensive expression profile along nephron segments [42]. The renal EC predominantly expressed three isoforms of Igfbp: *Igfbp3* (DVR), *Igfbp4* (AA), and *Igfbp5* (GEC/PCEC) (Figure 2E). *Igf1* and *Igf2* were mainly expressed by VEC and AA, respectively, and their receptors *Igf1r* and *Igf2r* were mainly expressed in a small proportion of ECs. This zone-distinct expression likely fine-tunes the segment-specific regulation of IGF signaling.

### 2.4. Transcriptional Responses in Progression of DKD

The diabetic phenotype for BTBR*ob*/*ob* mice is shown in Figure 1C,D and Appendix A. BTBR*ob*/*ob* mice showed elevated blood glucose and albuminuria levels (Figure 1C,D) with an increased glomerular size already evident at 6 weeks (Appendix A), confirming a DKD phenotype. Capillary density progressively declined over time in the BTBR*ob*/*ob* mice (Appendix A). In agreement, the FACS of PECAM1^+^ cells showed reduced numbers of ECs in 11- and 20-week-old diabetic mice (Figure 1B).

The UMAP showed that the majority of clusters for BTBRLean and BTBR*ob*/*ob* mice overlapped (Figure 3A). For s comparison of DEGs between non-diabetic and diabetic mice, PCEC/PCEC^IEG-high^ were pooled to PCEC, and GEC/GEC^IEG-high^ to GEC (Figure 3B, Appendix A). GEC and PCEC comprised the majority of captured cells and displayed the largest number of DEGs (Appendix A), and hence became the focus for further analysis. Strikingly, many overlapping DEGs changed the direction of regulation at 20 weeks compared to those at 6/11 weeks (Figure 3). Among overlapping genes, *Ucp2* and *Serinc3* were regulated similarly between PCEC and GEC. Mitochondrial Ucp2 reduces oxidative stress, and plasma membrane Serinc3 inhibits apoptosis [43].

### 2.5. Pathway and Network Analyses of EC Responses in DKD

We then performed IPA on PCEC and GEC DEGs to identify significantly enriched and regulated canonical pathways. In PCEC, multiple pathways were inactivated in the 11-week BTBR*ob*/*ob* mice contrary to at 6 weeks (Figure 4A, Appendix A), in conjunction with capillary rarefaction (Appendix A), and most pathways remained inactivated at 20 weeks. In contrast, GEC showed transient activations of multiple pathways at 6 weeks (Figure 5A, Appendix A), in conjunction with an increased glomerular area (Appendix A).

Oxidative phosphorylation (OXPHOS) and EIF2 signaling were among the top regulated pathways in GEC and PCEC. The inactivation of OXPHOS in PCEC started at 11 weeks in diabetic mice and was evident in both PCEC and GEC at 20 weeks (Figure 5A, Appendix A). Identified DEGs (Appendix A) at 20 weeks are shown in Appendix A. EIF2 signaling is central to eukaryotic translation initiation, and its dysregulation was manifested with a myriad of DEGs (Appendix A) and regulated functional complexes (Appendix A). PCEC and GEC displayed the opposite regulation of EIF2 signaling and upstream growth factor signaling at 6 weeks. In contrast, EIF2 signaling was inactivated in both PCEC and GEC at 20 weeks, concurrent with increased downstream apoptosis (Figure 4A and Figure 5A).

The IPA summary illustrates important pathways, effects, and regulators in PCEC (Figure 4) and GEC (Figure 5), with IGF signaling as a central factor. Intriguingly, while the IGF1 pathway was activated in GECs at both 6 and 20 weeks, it was inactivated in PCEC at 11 weeks. The analysis showed that *Igf1* and several *Igfbp* isoforms were altered in the GEC and PCEC from diabetic mice (Appendix A). *Igfbp5* was significantly regulated in GEC (Appendix A). *Igfbp5* RNA in situ hybridization showed increased expression in GEC and mesangial cells (Appendix A).

### 2.6. Alternative Splicing in Progression of DKD

The full-length sequencing data generated using SmartSeq2 allowed for an analysis of the differential splicing events (DSEs). The analysis distinguished five basic modes of splicing events (Figure 6A) and calculated the inclusion levels of exons or introns for each event. The median inclusion level difference for splicing events (FDR < 0.05) was ~0.4 between BTBRLean and BTBR *ob*/*ob* (GEC/PCEC) at each timepoint (Appendix A). We identified 903 (PCEC 11 wk) to 1111 (GEC 6 wk) significant DSEs (Figure 6B, DSE details and links to Sashimi plot are shown in Appendix A). The dominant AS mode was exon skipping, accounting for 60% of the overall DSEs, and about 29% of the DSE genes involved multiple DSEs (Appendix A).

For GEC and PCEC, 109 and 96 genes, respectively, involved DSEs at all timepoints (Figure 6C, Appendix A). As examples, *Lias*, *Chek2*, and *Tbc1d31* (Figure 6D,E) showed no differential gene expressions, but significant DSEs at all timepoints either at the same (*Lias* and *Chek2*/GEC) or different (*Tbc1d31*/PCEC) splice junctions were observed. *Lias*, essential for mitochondrial energy metabolism and redox regulation in DKD [44,45], displayed exon skipping in BTBRLean but not in BTBR*ob*/*ob* GEC. In contrast, *Chek2*, involved in DNA repair and cell cycle arrest upon DNA damage, displayed induced exon skipping in BTBR*ob*/*ob* GEC (Figure 6D). *Tbc1d31*, essential for ciliogenesis, showed reduced exon skipping at different splice junctions between 6- and 11/20-week-old BTBR*ob*/*ob* mice. Overall, the distinct and overlapping DSEs and involvement of different AS modes or junctions suggest an intricate regulation of AS in DKD progression.

### 2.7. DEG and DSE Linked to Divergent Biological Processes

To explore the relationship between differential transcription and splicing, we compared DEGs and DSE genes and found little overlap between these two regulatory mechanisms (Figure 7A). For DSE genes, enriched gene ontology biological processes (GOBPs) were predominantly related to DNA repair, epigenetics, and post-transcriptional modification, initiated already at 6 weeks (Figure 7B). Additionally, post-transcriptional processes, including RNA modification, processing, and splicing, were enriched in both EC subtypes across timepoints. Furthermore, GEC and PCEC displayed diversely enriched GOBPs, e.g., cilium assembly in PCEC and autophagy in GEC.

In contrast to DSEs, the top enriched GOBPs in DEGs were related to ribosomal complex and protein translation, more prominent at 11/20 weeks (Figure 7C). In comparison with DSE genes, DEGs were involved in more diverse biological processes. DEG-enriched GOBPs showed a more dynamic pattern, in accordance with the fewer overlapping DEGs between timepoints (Figure 3C,D). RNA-binding proteins (RBPs) regulate AS [46]. We identified 91 RBPs that were differentially expressed in GEC/PCEC for at least one timepoint. Most RBPs showed differential expression at 6 wees in GEC, but at 11 weeks in PCEC (Appendix A). Some of these RBPs are known to regulate splicing, e.g., *Rbm4b* [46] at 6 weeks, *Son* [47] at 11/20 weeks, and *Igf2bp2* [48] at 6/20 weeks.

Taken together, these results suggest divergent roles of DEGs and DSE genes linked through differentially expressed RBPs (Figure 7C). A proposed working model for DEGs and DSEs and related biological processes is shown in Figure 7D.

## 3. Discussion

To determine vascular heterogeneity in healthy kidneys and in the progression of DKD, we employed scRNAseq using the SmartSeq2 protocol [49,50]. In healthy kidneys, our data pinpointed EC populations similar to those described previously [3,4] but distinguished them more accurately, including AVR and VEC [3] as well as tip and proliferating ECs [4]. This is likely thanks to the superior mRNA capture and sequencing depth of SmartSeq2 compared to droplet-based methods. Our scRNAseq data identified, on average, >4400 genes/cell, compared with <2500 genes/cell in previous studies [3,4], thereby permitting the identification of novel EC population-specific markers, such as *Il33* in VEC/PCEC/AVR and *Ace* in AA. A pathway analysis of EC subtypes revealed extensive zonation of the immune system, coagulation system, vascular tone, and calcium homeostasis. We also observed the EC zonation of IGF signaling, underscored by previous studies [3]. These pathways may all account for the EC functions in DKD progression, and our data could facilitate therapeutic designs and the targeting of specific EC compartments.

To elucidate longitudinal changes in ECs during DKD progression, we studied three timepoints in the BTBR*ob*/*ob* mouse model that resemble early human DKD [14,51,52]. The absence of hypertension in this model allows the identification of the mediators of diabetic kidney disease independent of the confounding effects of coexisting hypertension, as may occur in other models [14]. Capillary rarefaction was initiated as early as 6 weeks in the BTBR*ob*/*ob* mice. Our data suggest an opposite regulation of many genes over time in diabetes. A similar pattern has been observed in whole-kidney transcriptomics from patient biopsies, suggesting compensatory and adaptive mechanisms [53].

The IPA on DEGs for PCEC and GEC in our study found significantly enriched and regulated canonical pathways in DKD progression. The inactivation of OXPHOS in PCEC and GEC at 20 weeks is likely associated with GEC dysfunction and PCEC loss, as OXPHOS and mitochondrial dysfunction are linked to DKD and vascular rarefaction [54,55,56]. The activation of EIF2 in GEC is likely caused by compensatory glomerular growth factor signaling in DKD [57]. The inactivation in PCEC possibly represents an adaptive response to cellular stress through the inhibition of global translation [58]. Moreover, the persistent inactivation of EIF2 signaling may be associated with apoptosis, which is common in both EC populations at late timepoints.

IGF1 signaling was a central factor altered in diabetic mice. Interestingly, GEC and PCEC showed contrariwise changes: activation in GEC and inactivation in PCEC. The IGFBP family fine-tunes IGF1 signaling, displaying distinct spatial expressions and functions [59]. *Igfbp5* was expressed in both GEC and PCEC and was significantly regulated in GEC in diabetes, as shown in human DKD [28], and enriched in GEC/PCEC in human kidney [27,29]. These observations implicate IGFBP5/IGF1 in the pathogenesis of DKD, and thus emphasize the functional significance of our observation of IGF1-related DEGs.

We performed the first comprehensive profiling of the AS landscape in renal ECs during DKD progression, which may shed light on the intricate regulation of the kidney disease evolution. Aberrant AS has been implicated in chronic kidney diseases [46], diabetes [60], and diabetic vasculopathy [61,62], where multiple splice isoforms exhibit functional significance during disease progression. Notably, AS can serve as a therapeutic target [46], given that the protective effects of the SGLT2 inhibitor are partially mediated by the regulation of AS in the proximal tubule in DKD [9]. The DSE profile in this study provides a comprehensive data source for interrogating splice isoform switching in early DKD and their biological relevance. DNA repair emerged as a key biological process associated with aberrant AS. DNA damage and repair are implicated in the pathogenesis of DKD, as manifested in urine-derived cells in diabetic patients [63] and in podocytes in the DKD model [64].

Our study has certain limitations. First, only female BTBR mice were included in the study due to the rapid disease progression and severe phenotypes in male BTBR*ob*/*ob* mice causing loss of animals in breeding. Potential gender-specific transcriptomic changes were not studied. Second, we cannot rule out the possible loss of Pecam1 protein in individual ECs during DKD progression, which could lower their representation in the scRNAseq data. Third, the validation of DEGs and splice isoform switching in scRNA-seq data remains challenging. While we can reliably annotate individual clusters to the correct EC subtype via IHC and RNA-ISH, it was not possible to use IHC, RNA-ISH, or RT-PCR to validate quantitative differences between different EC subpopulations. Fourth, GOBPs were used to compare DEGs and DSEs, as IPA is not fitted for splice isoform analyses. Fifth, the SmartSeq2 protocol has an advantage in the context of alternative splicing analysis, since it mitigates the 3′ bias intrinsic to, e.g., droplet-based technologies, and results in full-length transcript coverage. However, the absence of unique molecular identifiers (UMIs) makes this approach more sensitive to PCR amplification bias, which could hinder a correct assessment of the alternative splicing [65]. Moreover, it is a general limitation for scRNA-seq that the low amount of starting material together with transcriptional bursting results in a sparse count matrix with dropouts that cannot easily be distinguished from true biological absences of gene expressions. Quantification at the transcript level is even more sensitive to this effect. While SmartSeq2 mitigates this limitation by enabling deeper sequencing per cell, it does, on the other hand, not have high throughput and is thus limited in the number of cells in total and per cluster, with reduced power as a result.

In conclusion, we built a single-cell renal EC transcriptomic atlas and identified novel EC markers that characterize different EC populations along the intrarenal microvasculature. We discovered the differential regulation of GEC/PCEC during the progression of murine DKD. We found that the inactivation of OXPHOS in PCEC and GEC is likely associated with GEC dysfunction and PCEC loss. Additionally, the activation of EIF2 in GEC may be linked to compensatory glomerular growth factor signaling in DKD while inactivation in PCEC likely represents an adaptive response to cellular stress. Moreover, the involvement of differential splicing and transcription in divergent biological processes is novel data and adds important information to our atlas. The information provided by our study will allow for more a precise targeting of intrarenal microcirculation during disease progression and, in turn, the discovery of therapeutic targets.

## 4. Materials and Methods

Kidney ECs were collected from diabetic BTBR*ob*/*ob* mice and non-diabetic littermates at different timepoints. Their heterogeneity and changes in transcription and alternative splicing during DKD progression were mapped using SmartSeq2 scRNAseq and elucidated through pathway, network, and gene ontology enrichment analyses. Identified EC populations were validated via immunohistochemistry and in situ hybridization.

### 4.1. Animals

All animal experiments were carried out in accordance with Swedish legislation and local guidelines and regulation for animal welfare and were approved by the Regional Laboratory Animal Ethics Committee of Gothenburg, Sweden, (ID 38-2015) and the Regional Laboratory Animal Ethics Committee of Uppsala, Sweden, (ID C110-12 and 5.8.18-04862-220). In this study, we used several mouse strains, all maintained as breeding colonies at a local animal facility. All animals were housed with a 12 h light–12 h dark cycle and had ad libitum access to water and chow. The housing temperature was kept at 20 ± 2 °C, and the relative humidity, at 50 ± 5%. BTBRV(B6)-Lepob/WiscJ stock no. 004824 (Jackson Laboratory, Bar Harbor, ME, USA) female diabetic BTBR*ob*/*ob* mice, a model of diabetic nephropathy, and their non-diabetic littermates, heterozygous or wildtype mice (BTBRLean), were euthanized, and their kidneys harvested, at 6, 11, and 20 weeks of age (Figure 1A). Three mice per genotype per age group were randomly selected from the breeding cohort to be included in the single-cell analysis. Blood glucose levels and the urinary albumin–creatinine ratio (UACR) were measured at 11 weeks of age in the BTBRLean (*n* = 9) and BTBR*ob/ob* (*n* = 11) mice. Capillary rarefaction and glomerular size were evaluated in 6-, 11-, 15-, and 20-week-old BTBRLean and BTBR*ob/ob* mice (*n* = 3–7 mice, as indicate by dot blots of data). For the validation of EC clusters, we used 1 male and 2 female 6–20 week-old C57Bl6/J (Jackson Laboratory) mice and reporter mice with Cldn5^GFP^ (Tg(Cldn5-GFP)Cbet/U) (2 females) or Cdh5-TdTomato (1 male, 2 females) on a C57BL6/J background. Cdh5-TdTomato mice were inducible Cdh5-Cre^ERT2^ mice [66] bred to a reporter mouse; Ai14-TdTomato mice [67] were induced with three doses of tamoxifen (2 mg) in peanut oil via oral gavage at 4 weeks of age.

### 4.2. Isolation of Renal Single Cells

Mice were euthanized via cervical dislocation, and their kidneys were immediately harvested and placed into an ambient phosphate-buffered saline (PBS) solution (DPBS, Thermo Fisher Scientific, Dreieich, Germany). The papilla was removed, and the tissue was then diced and incubated in dissociation buffer, 0.13 U/mL Liberase TL (5401020001, Merck, Darmstadt, Germany) in RPMI 1640 (Thermo Fisher Scientific) at 37 °C in a water bath with horizontal shaking of 500–800 rpm for 20–30 min. Thereafter, the cell suspension was sequentially passed through 70 µm and 40 µm cell strainers. The 70 µm cell strainer was additionally washed with 5 mL RPMI 1640. Cells were pelleted via centrifugation at 300× *g* for 15 min at 4 °C, with the supernatant then removed, and the pellet resuspended in ACK lysing buffer (A1049201, Thermo Fisher Scientific) for 5 min. After the lysing of red blood cells, the cells were pelleted via centrifugation at 300× *g* for 10 min at 4 °C; supernatant, removed; and pellet, resuspended in FACS buffer (DMEM supplemented with 1% fetal calf serum and 5 mM EDTA). For the labeling of EC and live cells, the cell suspension was incubated with APC-conjugated anti-Pecam1 antibody (1:100, 551262, BD Biosciences, Heidelberg, Germany) and Calcein AM (2 µM, C3099, Thermo Fisher Scientific) for 20 min at room temperature (RT), and then centrifugated as above; supernatant, removed; and the cell pellet, resuspended in FACS buffer and placed on ice.

### 4.3. Fluorescence-Activated Cell Sorting (FACS)

Antibody-stained cell suspensions were analyzed using a Beckton Dickinson FACSAria III cell sorter equipped with a 100 µm nozzle. Single cells meeting the selection criteria as described below were deposited into 384-well plates containing 2.3 µL lysis buffer (0.2% Triton X-100, 2 U/mL RNase inhibitor, 2 mM dNTPs, 1 µM Smart-dT30 VN, ERCC 1:4 × 10^4^ dilution). For single-cell sorting, first, a gate for the forward scatter area/side scatter area (FCS-A/SSC-A, linear scale) was set generously around present events, excluding events only with low values (cell debris and red blood cells). Second, doublet discrimination was implemented using the FCS-A/FSC-height and SSC-A/SCC-height. Third, selected events were then analyzed for live cell markers and positive fluorescent signals; unstained cells were used to ensure correct gating and as negative controls. Plates were briefly centrifugated prior to sorting, while the correct deposition of the droplet into the 384-well plate was controlled by test spotting beads onto the seal of the respective plate, and if necessary, the plate position was adjusted for each new plate placed in the machine. The sample and plate holder of the machine were kept at 4 °C, and the plates were placed on dry ice immediately after the sorting was completed and subsequently stored at −80 °C until downstream processing.

### 4.4. SmartSeq2 Library Preparation and Sequencing

Single-cell cDNA libraries were prepared according to a previously described SmartSeq2 protocol [68]. In brief, mRNA was transcribed into cDNA using the oligo(dT) primer and Superscript II reverse transcriptase (Thermo Fisher Scientific). Second-strand cDNA was synthesized using a template-switching oligo. The synthesized cDNA was then amplified via polymerase chain reaction (PCR) for 24 cycles. Purified cDNA underwent quality control (QC) through analysis on a 2100 Bioanalyzer with a DNA high-sensitivity chip (Agilent Biotechnologies, Santa Clara, CA, USA). When the sample passed the QC, the cDNA was fragmented and tagged (tagmented) using Tn5 transposase, and each single cell was uniquely indexed using the Illumina Nextera XT index kits (Set A-D). Thereafter, the uniquely indexed cDNA libraries from one 384-well plate were pooled into one sample to be sequenced on one lane of a HiSeq3000 sequencer (Illumina, San Diego, CA, USA), using the sequencing strategy of dual indexing and single 50 base-pair reads.

### 4.5. Sequence Data Processing

Pooled single-cell cDNA library samples were sequenced as described above. Demultiplexing into single-cell fastq files was performed with the standard parameters of the Illumina pipeline (*bcl2fastq*) using Nextera index adapters. The demultiplexed individual fastq files were then aligned to the mouse reference genome GRCm38 (mm10), using TopHat (version 2.1.1) [69,70], and the adapter sequences in each read were removed using trim galore before read mapping. The alignment BAM files were sorted according to the mapping position, and the duplicated reads were removed using Samtools software (version 0.1.18). The raw read counts for each gene were calculated using featureCounts from the Subread package (version 1.4.6-p5) [71]. Cells with total read counts less than 50,000 were removed. Cells that displayed a clearly contaminated transcriptome with, e.g., an epithelial cell-specific gene signature (*Lrp2*, *Pax8*, *Slc12a1*, and *Slc12a3*) were excluded. This resulted in a final dataset for analysis composed of cells from 3 lean and 3 obese female mice from most timepoints (6, 11, and 20 weeks), with the exception of the 6-week lean mice (*n* = 2, one sample removed due to poor quality). After sequencing and quality control, a dataset of 5464 cells was constructed and used for downstream bioinformatic analyses.

The dataset was imported into the R-Seurat package for clustering, visualization, and marker gene analysis (version: 3.1.1) [72]. For the visualization of all single cells, the dimension reduction method UMAP was applied (UMAP: uniform manifold approximation and projection) [73]. The count values were normalized to a 500,000 count library size per cell for visualization as bar plots. Differentially expressed gene (DEG) analysis was performed using the FindMarkers function in the Seurat package. Single-cell ECs from non-diabetic (BTBRLean) mice were used to identify DEGs between EC compartments in the dataset, using cutoff values of foldchange >1.5 and an adjusted *p*-value < 0.05. DEGs were identified in the same way as above to compare diabetic ECs (BTBR*ob*/*ob*) and non-diabetic ECs (BTBRLean) for peritubular capillary endothelium (PCEC) and glomerular endothelium (GEC) at the different timepoints. For this analysis, cells were pooled regardless of their immediate early gene transcripts (IEGs): PCEC and PCEC^IEG-high^ were pooled to PCEC, and GEC and GEC^IEG-high^ were pooled to GEC. While all mice were female (confirmed through external genitalia), 2 mice showed expressions of X-chromosome inactivation genes (*Tsix* and *Xist*). These genes were removed from the heatmap visualizations but included in the calculations of the DEGs (Appendix A).

A published single-cell dataset [27] was used for comparisons with human GEC and PCEC. In these data, we annotated cluster N4 (*GATA5*+, *EHD3*+) as GEC and cluster N2 as PCEC, as previously described [3,4], which were then compared with our mouse GEC and PCEC, respectively.

GWAS (genome-wide association study) genes associated with CKD [28] were tested for expression in our dataset and presented in a heatmap of EC populations.

### 4.6. Ingenuity Pathway Analysis

Ingenuity Pathway Analysis (IPA version 111725566, Qiagen, Hilden, Germany) was utilized to describe pathways and their regulation by DEGs in our dataset. All IPAs conducted had criteria of *p* < 0.05 and z score > 2. Based on the function and expression of the DEGs, the IPAs also assigned an activation score to each enriched pathway, which were further compared between all EC clusters using IPA comparison analysis. In addition, IPA predicts which upstream regulators are activated or inhibited to explain the up-regulated and down-regulated genes observed in the dataset, while downstream effect analysis enables the visualization of biological trends in the experiment and predicts the effects of molecular changes observed in the dataset on biological processes and diseases or on toxicological functions. An IPA graphical summary illustrates the most important pathways, effects, and regulators.

### 4.7. Alternative Splicing Analysis

Raw reads were aligned to the mouse reference genome GRCm38 (mm10) using STARSolo (STAR v2.7.5c) [74,75]. Cells of each group of interest were combined into pseudo-bulk samples by merging bam files of the individual cells into one bam file per group using Samtools (v1.15.1) [76]. Differential alternative splicing analysis between different pairwise group comparisons was performed with rMATS (v4.1.2) [77] using the merged bam files as input. rMATS distinguished five basic modes of splicing events and calculates the inclusion levels of exons or introns for each event. This generated five result tables, one for every splicing event mode (SE, MXE, RI, A5SS, A3SS), containing the *p*-value, FDR, and inclusion level difference for each comparison. For further analysis, the result files based on both junction counts and exon counts were used. Sashimi plots for the selected splicing events were generated using rmats2sashimiplot (v2.0.4).

### 4.8. Gene Ontology Enrichment Analysis

GO-term enrichment analysis was run using the enrichGO function of the R package ClusterProfiler (v4.2.0) [78]. For the GO-term enrichment of the DSEs, genes involving significant differential splicing events (FDR < 0.05) and an absolute inclusion level difference >0.4 were used as the input gene list [78]. For the GO-term enrichment of the DEGs, significantly (adjusted *p* < 0.05) differentially expressed genes were used as the input gene list. Heatmaps were created for the top 10 GO terms for each comparison using the R package pheatmap (v1.0.12) (https://CRAN.R-project.org/package=pheatmap).

### 4.9. Identification of RNA Binding Proteins

DEGs were compared with a compiled list of mouse RBPs from 3 databases, rMAPS (http://rmaps.cecsresearch.org, accessed on 12 October 2022), RBPDB (http://rbpdb.ccbr.utoronto.ca, accessed on 12 October 2022), and Splicesome (http://spliceosomedb.ucsc.edu, accessed on 12 October 2022). The list consisted of 659 unique RBPs in total.

### 4.10. Validation of EC Cluster Markers with RNA In Situ Hybridization (RNA-ISH)

For RNA-ISH, the RNAscope*^®^* system (Advanced Cell Technologies, Worcester, MA, USA) was applied according to the manufacturer’s protocol. In brief, tissue sections were prepared as described below (Immunofluorescence staining). After dehydration, HRP was quenched with Bloxall (Sp-6000, Vector Technologies, Stuttgart, Germany) blocking solution for 10 min at RT followed by Pretreat III solution for 30 min at RT. Then, RNAscope^®^probes (Appendix A) were hybridized on the sections for 2 h at 40 °C, and then underwent an RNAscope^®^ multiplex fluorescent Reagent kit v2 assay according to the manufacturer’s instructions. Opal 520, Opal 570, and Opal 690 (Akoya Biosciences, Marlborough, MA, USA) were diluted 1:1000. Sections were mounted with ProLong^®^Gold mounting medium, and images were obtained using a confocal microscope (Leica Microsystems Sp8, Wetzlar, Germany) at 400×.

### 4.11. Immunofluorescence Staining

Standard methods for immunostaining were used. In brief, the harvested kidneys were fixated in 4% formaldehyde for 4 h at RT, followed by immersion in 30% sucrose/PBS solution for at least 24 h at 4 °C. The tissues were then embedded for cryo-sectioning in Tissue-Tek OCT and sectioned at 14 µm on a Cryostat NX70 (Thermo Fisher Scientific). The sections were stored at −80 °C. For staining, sections were blocked >1 h at RT with blocking buffer (X0909, Dako, Agilent) supplemented with 0.25% Triton X-100 (Sigma Aldrich, St. Gallen, Switzerland). Thereafter, the sections were sequentially incubated with primary antibodies overnight at 4 °C and corresponding fluorescently conjugated secondary antibodies (Appendix A) and Hoechst 33342 (Dako) for 2 h at RT. The sections were mounted with ProLong^®^Gold mounting medium (Thermo Fisher Scientific). Images were analyzed utilizing ImageJ version 1.54 (NIH) software [79]. For the quantification of the capillary density, the endothelial marker Pecam1 was quantified via immunohistochemistry and light microscopy in paraffin sections of kidneys from 5 BTBR*ob*/*ob* and 5 BTBR Lean 11-week-old mice for different regions of the kidney: cortex, outer stripe of the medulla (OSOM), inner stripe of the medulla (ISOM), and inner medulla (IM). Large arteries and glomeruli were excluded. To document changes in capillary density over time, we performed immunofluorescence and confocal imaging on the renal cortex with additional endothelial markers, endomucin and podocalyxin (3 BTBR*ob*/*ob* and 3 BTBR Lean mice for each timepoint). The stained area relative to the total area was calculated using Otsu thresholding from 5 images from the cortex of each mouse.

### 4.12. Blood and Urine Analysis

Non-anesthetized plasma glucose levels were measured in the tail vein blood at 11 weeks of age using a portable glucometer (Accu-Chek mobile^®^, Solna, Sweden) following the manufacturer’s instructions. The urinary albumin–creatinine ratio (UACR) was measured in spot urine collected on a tray. The urine albumin–concentration was measured with ELISA (Albuwell-M), and the urine creatinine was measured using a colorimetric/fluorometric assay kit (ab65340 Creatinine Assay Kit, Abcam) following the manufacturer’s instructions.

### 4.13. Statistical Analysis

For capillary density measurements, data were expressed as mean ± SD. Means between groups were compared using a 2-tailed unpaired Student’s *t*-test or one-way ANOVA with Bonferroni’s multiple comparisons, where appropriate, using GraphPad Prism version-8 (GraphPad Software Inc., La Jolla, CA, USA). To identify DEGs, the individual cells between groups were compared using the Wilcoxon Rank Sum test. Multiple-test correction was performed using the Bonferroni method, and the corrected *p* value < 0.05 was set as significant. All data were tested for uneven distribution, and in the case of an uneven distribution, logarithmic values were used. *p* < 0.05 was considered statistically significant.

## Figures and Tables

**Figure 1 ijms-25-04320-f001:**
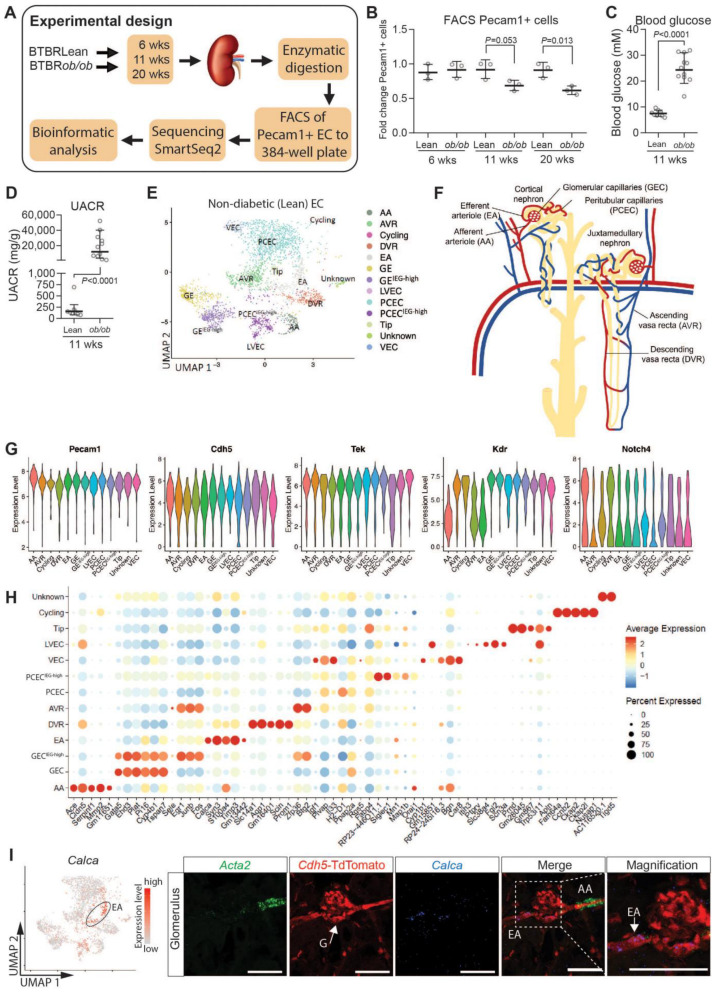
Experimental design and EC populations. (**A**) Overview of the experimental design. (**B**) Kidneys were harvested and dissociated into single-cell suspensions before labeling with Pecam1 antibody to sort endothelial cells into 384-well plates. (**C**) Blood glucose levels and (**D**) urinary albumin–creatinine ratios (UACRs) in 11-week-old non-diabetic (BTBRLean) and diabetic (BTBR*ob*/*ob*) mice. (**E**) Single-cell sequencing utilized SmartSeq2 technology, and data were subjected to bioinformatic analysis. UMAPs of endothelial cells from non-diabetic (BTBRLean) mice were used to identify 13 separate clusters of EC annotated as afferent arteriole/arterioles (AA), ascending vasa recta (AVR), proliferating (cycling), descending vasa recta (DVR), efferent arteriole (EA), glomerulus (GEC), lymphatics (LVEC), peritubular capillaries (PCEC), tip cells (tip), and veins (VEC). GEC and PCEC each had two clusters with one cluster displaying a high immediate early gene expression (GEC^IEG-high^ and PCEC^IEG-high^). (**F**) Schematic of renal EC populations. (**G**) Violin plots showing pan EC markers including *Pecam1*, *Cdh5*, *Tek*, *Kdr*, and *Notch4* in BTBRLean mice (**H**). Dot blot of markers for each EC population in BTBRLean mice. (**I**) *Calca* expression in efferent arteriole of C57BL6/J mice. Scale bars = 50 µm. Data expressed as mean ± SD. Other EC population validations are shown in Appendix A.

**Figure 2 ijms-25-04320-f002:**
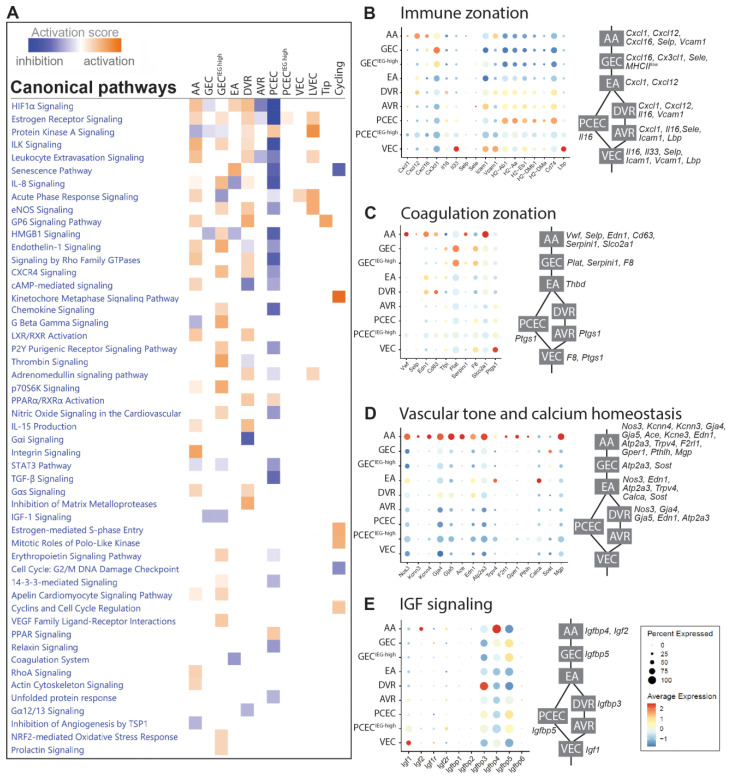
Heterogeneity of different EC compartments in BTBRLean mice. (**A**) Selection of significant canonical pathways identified through pathway analysis based on DEGs between EC populations in non-diabetic (Lean) mice (complete IPA results shown in Appendix A). Dot blot and schematic representation of EC populations for genes involved in (**B**) immune zonation, (**C**) coagulation zonation, (**D**) vascular tone and calcium homeostasis, and (**E**) IGF signaling.

**Figure 3 ijms-25-04320-f003:**
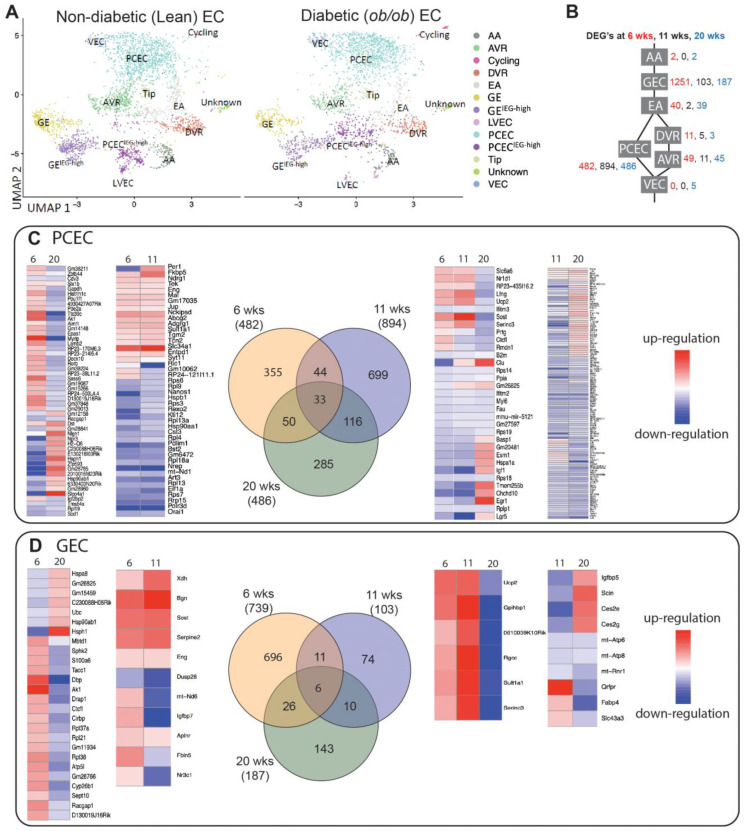
Renal EC response to DKD progression in BTBR*ob*/*ob* mice. (**A**) UMAP of endothelial cells in non-diabetic (Lean) and diabetic(*ob*/*ob*) mice. (**B**) Schematic of DEGs of different EC populations in non-diabetic and diabetic mice at 6, 11, and 20 weeks. (**C**,**D**) Venn diagrams of DEGs for PCEC and GEC, respectively, comparing non-diabetic (Lean) with diabetic (*ob*/*ob*) mice at different timepoints. Expression heatmaps (blue, low; red, high) of the relative expression are shown for each comparison.

**Figure 4 ijms-25-04320-f004:**
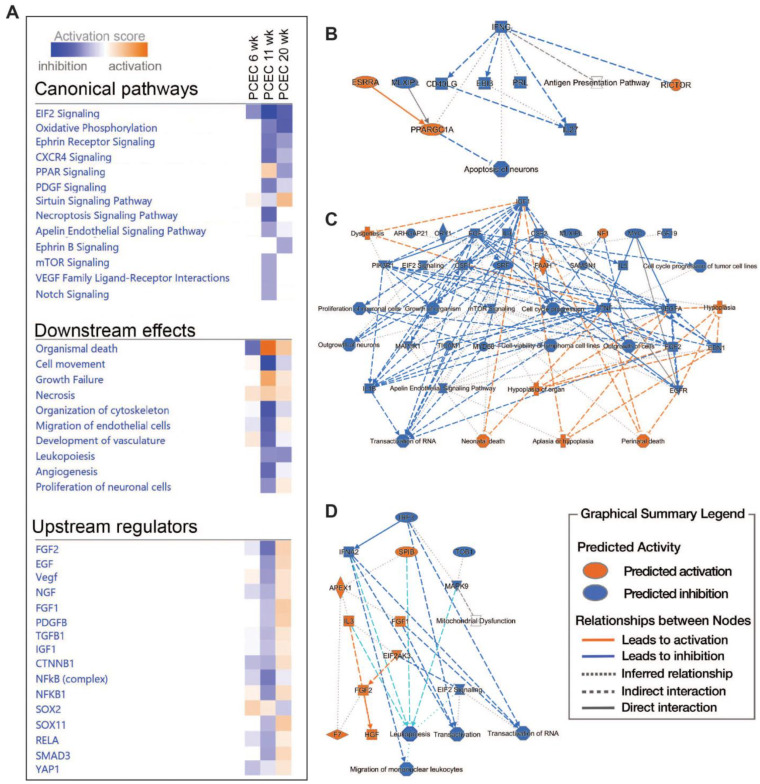
Pathway analysis (IPA) of PCEC based on DEGs comparing non-diabetic (Lean) and diabetic (*ob*/*ob*) mice. (**A**) The selection of significant canonical pathways, downstream effects, and upstream regulators identified through IPA for PCEC (the complete IPA results are shown in Appendix A). (**B**–**D**) Graphical summaries of pathway analysis showing the pathway activation (red) or pathway inhibition (blue) in PCEC in diabetic 6-, 11-, and 20-week-old mice, respectively.

**Figure 5 ijms-25-04320-f005:**
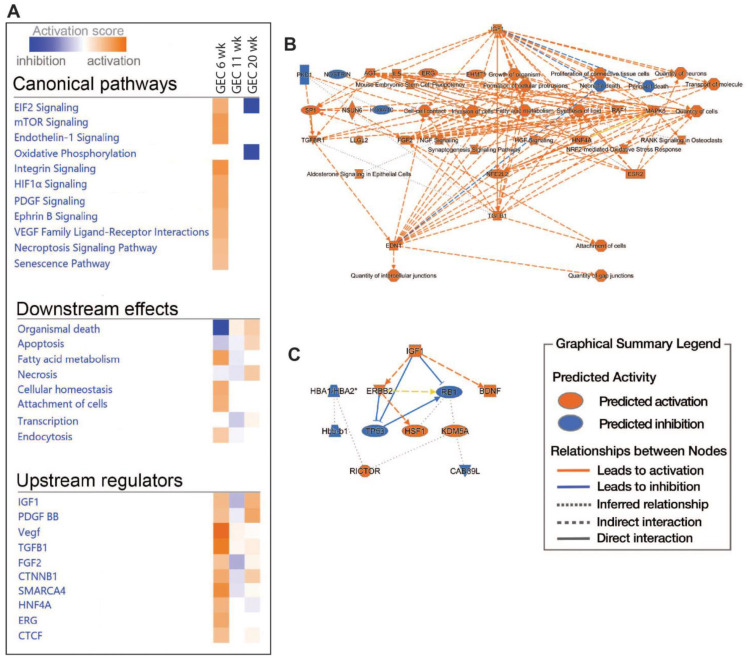
Pathway analysis (IPA) of GEC based on DEGs comparing non-diabetic (Lean) and diabetic (*ob*/*ob*) mice. (**A**) The selection of significant canonical pathways, downstream effects, and upstream regulators identified through IPA for GEC. The complete IPA results can be found in Appendix A. (**B**,**C**) Graphical summaries of the pathway analysis showing activation (red) or inhibition (blue) in diabetic GEC in 6- and 20-week-old mice, respectively. No significant pathway regulation was found in 11-week-old mice. * multiple identifiers in the dataset file map to a single gene.

**Figure 6 ijms-25-04320-f006:**
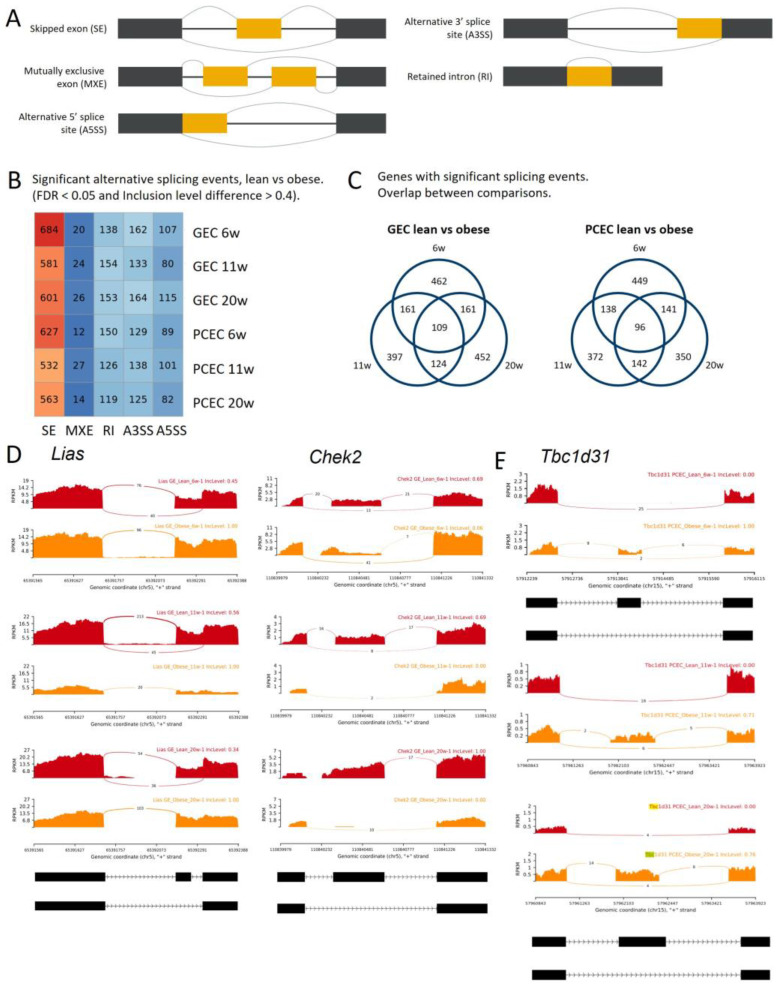
Identification of DSEs in DKD progression in BTBR*ob*/*ob* mice. (**A**) Schematics of five different modes of AS distinguished through rMATS. (**B**) Number of significant DSEs identified in comparison between BTBRLean and BTBR*ob*/*ob* GEC and PCEC at different timepoints. (**C**) Venn diagrams of DSE genes for PCEC and GEC, respectively, displaying number of overlapping or distinct DSE genes at different timepoints. Exemplary Sashimi plots visualizing read counts of splice junctions and inclusion levels of exons at selected genomic locations for *Lias* and *Chek2* in GEC (**D**) and *Tbc1d31* in PCEC (**E**) in BTBRLean and BTBR*ob*/*ob* mice.

**Figure 7 ijms-25-04320-f007:**
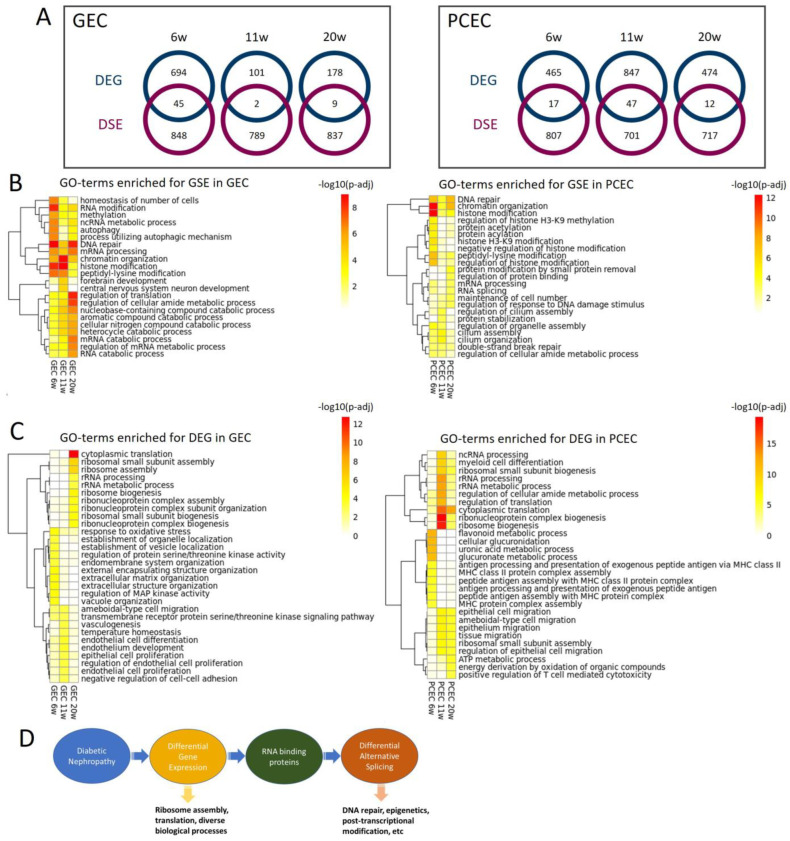
Differentially related biological processes with DSE genes and DEGs. (**A**) Venn diagrams of DSE genes and DEGs for PCEC and GEC, displaying number of distinct genes and overlapping genes that are both differentially expressed and alternatively spliced. GO-term enrichment analysis for DSE genes (**B**) and DEGs (**C**) displaying top 10 significant GOBPs at three timepoints in GEC and PCEC, respectively. Scale bar represents -log_10_ (adjusted *p*-value). (**D**) Proposed working model displaying the link between DEGs and DSEs and their divergently regulated biological processes.

## Data Availability

ScRNA-seq data can be accessed from NCBI’s Gene Expression Omnibus database, accession number GSE192687. All data used to support the findings of this study are included in the paper and the Appendix A and are available from a searchable database at https://betsholtzlab.org/Publications/MouseKidenyOB/database.html.

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
