# Peer review of "Renal Endothelial Single-Cell Transcriptomics Reveals Spatiotemporal Regulation and Divergent Roles of Differential Gene Transcription and Alternative Splicing in Murine Diabetic Nephropathy"

_ijms, 2024, doi:10.3390/ijms25084320_

Round 1

Reviewer 1 Report

Comments and Suggestions for Authors

1.     Endothelial cell transcriptomic correlation with clinical ( urine proteinuria & Serum creatinine) and renal histopathology needs to be shown in the results to substantiate the conclusions. 

2.     Endothelial cell changes in diabetic nephropathy like endothelial dysfunction , endothelial to mesenchymal transition evidenced by upregulation of VEGF, TGFbeta , BMP7 etc need to be shown in the results if data available. 

3.     Apart from the already known changes in the endothelial cells due to diabetic nephropathy any new pathways found in the study need to clearly explained in the results and also need to discuss the relevance in the discussion section. 

Author Response

Dear reviewer,

We appreciate your constructive comments. Please find below our response. Thank you.

  1. Endothelial cell transcriptomic correlation with clinical (urine proteinuria & Serum creatinine) and renal histopathology needs to be shown in the results to substantiate the conclusions.

We agree with the reviewer that it is interesting to have a correlation with disease progression to our data. The disease progression in BTBRob/ob mice with age is well established in previous reports. Our study also showed progressive glomerular hypertrophy and vascular rarefaction in different age groups (Suppl. Fig. S5). However, it is complex to run correlation analysis in single cell studies since these studies generate transcriptomic data of a number of cell types. Moreover, correlation analyses demand a relatively large animal cohort to generate reliable data, which majority of the single cell studies do not meet. Our study was more suited for differential gene expression analysis.

  1. Endothelial cell changes in diabetic nephropathy like endothelial dysfunction , endothelial to mesenchymal transition evidenced by upregulation of VEGF, TGFbeta , BMP7 etc need to be shown in the results if data available.

As the reviewer points out endothelial dysfunction is one of the hallmarks of DKD. We demonstrate for the first time in the current study that BTBRob/ob mice have progressive loss of peritubular capillaries (Suppl. Fig. S5F, G). Ingenuity pathway analysis shows significant regulation of canonical pathways (Suppl. File 2, page 2) in PCEC with several growth factors as upstream regulators (Suppl. File2, page 4). These include FGF2, PDGFB, TGFB1 which all show more activation across time points 6- 11- and 20- weeks of age. In contrast, VEGFA signalling is stronger at 6- and 20-weeks of age but weaker at 11-weeks. The same analysis was done in GEC (Suppl. Fil 2, page 8), with VEGFA signalling starting strong but declining over time. Our results highlight the importance of understanding signalling in different endothelial compartments.

  1. Apart from the already known changes in the endothelial cells due to diabetic nephropathy any new pathways found in the study need to clearly explained in the results and also need to discuss the relevance in the discussion section.

We agree with the reviewer that novel pathways should be highlighted in the results. The major novelty of our study is to reveal transcriptomic changes following disease progression in DN model, particularly in different EC subtypes. Even though multiple reported pathways in our study have been shown to be related to DN previously, our study provided a high level of granularity in terms of differential regulation of these pathways in different EC subtypes and different disease stages. We have found that the alteration of EIF2 signalling, oxidative phosphorylation, and IGF1 signalling are particularly striking and thus focused on these pathways in this study. On the other hand, to meet interests of a broad readers, we have listed all enriched pathways in the supplement file 2 and created a searchable database.

Your sincerely,

Pernille B. Laerkegaard Hansen, Ph.D.

Executive Director, Head of Bioscience Renal

Research and Early Development Cardiovascular, Renal and Metabolism,

BioPharmaceuticals R&D, AstraZeneca

Pepparedsleden 1, Gothenburg, 43183, Sweden

Reviewer 2 Report

Comments and Suggestions for Authors

The manuscript by Zhou et al. entitled "Renal endothelial single-cell transcriptomics reveals spatiotemporal regulation and divergent roles of differential gene transcription and alternative splicing in murine diabetic nephropathy” characterized spatiotemporal regulation of gene transcription and AS linked to DKD progression using SmartSeq2 single-cell RNA sequencing (scRNAseq). Overall, the experiments are expertly carried out and the paper is well-written. There are still some suggestions to improve the quality of the manuscript although the findings in this manuscript are potentially interesting.

1. Although bioinformatic data are clear, they did not provide wet-lab experiment to confirm the results. For example, RT-PCR should be performed in Figure 6.

2. I think comparison between DSE genes and DEGs is meaningless. That is different stage of gene regulation.

3. The limitation of analyzing alternative splicing using SmartSeq2 single-cell RNA sequencing should be discussed.

Author Response

Dear Reviewer,

We appreciate your constructive comments. Please find below our response. Thank you.

The manuscript by Zhou et al. entitled "Renal endothelial single-cell transcriptomics reveals spatiotemporal regulation and divergent roles of differential gene transcription and alternative splicing in murine diabetic nephropathy” characterized spatiotemporal regulation of gene transcription and AS linked to DKD progression using SmartSeq2 single-cell RNA sequencing (scRNAseq). Overall, the experiments are expertly carried out and the paper is well-written. There are still some suggestions to improve the quality of the manuscript although the findings in this manuscript are potentially interesting.

  1. Although bioinformatic data are clear, they did not provide wet-lab experiment to confirm the For example, RT-PCR should be performed in Figure 6.

We agree with the reviewer that lack of wet-lab validation in Figure 6 if a limitation of the study. We have now elaborated this limitation in the discussion. Tubular epithelial cells are predominant cell types in kidney, which adds a strong technical challenge to validate EC gene expression level by tissue RT-PCR. It is also very challenging to purify different EC subtypes with good surface markers and in a sufficient number for analyses. To clarify this, we have added the following section to Discussion: Second, the validation of DEGs in scRNA-seq data remains challenging. While we can reliably annotate individual clusters to the correct EC subtype by IHC and RNA-ISH, it was not possible to use IHC, RNA-ISH or RT-PCR to validate quantitative differences between different EC subpopulations.

  1. I think comparison between DSE genes and DEGs is meaningless. That is different stage of gene regulation.

We apologise if “comparison” in Figure 7 legend is misleading. Findings in this study revealed that different genes were involved in different stages of gene regulation. The figure legend is now changed to “Differentially related biological processes for DSE genes and DEGs”.  

  1. The limitation of analysing alternative splicing using SmartSeq2 single-cell RNA sequencing should be discussed.

We have added the section below to the Discussion to address this comment:

“SmartSeq2 protocol has an advantage in the context of alternative splicing analysis, since it mitigates the 3’ bias intrinsic to e.g. droplet-based technologies and results in full-length transcript coverage. However, the absence of unique molecular identifiers (UMIs) makes this approach more sensitive to PCR amplification bias, which could implicate correct assessment of alternative splicing (Arzalluz-Luque and Conesa, PMID: 30097058). Moreover, it is a general limitation for scRNA-seq as the low amount of starting material together with transcriptional bursting results in a sparse count matrix with drop-outs that cannot easily be distinguished from true biological absence of gene expression. While SmartSeq2 mitigates this limitation by enabling deeper sequencing per cell, it is on the other hand not high throughput, thus limited in number of cells in total and per cluster, with reduced power as a result.

Your sincerely,

Pernille B. Laerkegaard Hansen, Ph.D.

Executive Director, Head of Bioscience Renal

Research and Early Development Cardiovascular, Renal and Metabolism,

BioPharmaceuticals R&D, AstraZeneca

Pepparedsleden 1, Gothenburg, 43183, Sweden

Reviewer 3 Report

Comments and Suggestions for Authors

The authors studied temporal and spatial transcriptomics in various vascular compartments in a murine model of diabetic kidney disease. The paper is very interesting, but my initial enthusiasm was tempered by some methodological concerns and the presentation of results.

My main concerns:

1-The authors exclusively used female diabetic BTBR ob/ob and the authors compared their results with a previous study published by David Barry (Nature communications 2019) that was conducted in 4-month male mice. The authors should justify their choice of using only females and consider the hormonal cycle of female mice could potentially influence the results.

Additionally, the author mentioned that used another mice strain (C57Bl6/J) for validation purposes. However, the sex of mice used was not specified. It is important to justify the use of different strains for a validation experiment. Furthermore, the results of these experiments are not presented. The authors also used reporter mice with Cldn5GFP (Tg(Cldn5-GFP)Cbet/U) or Cdh5-TdTomato.  Was this control decided upon at the beginning of the study or was it included later?

2-The Number of mice used is not clearly provided, and there is no evaluation of the kidney lesions of diabetic or measurement of creatinine and albuminuria. This data should be included in the first figure.

3-Regarding the scRNA seq, the authors employed a pooling strategy that which may introduce challenges, such as potential cross-contamination between samples during library preparation or sequencing errors that could affect downstream analysis. Please, specify in the methods the quality control measures implemented to address these issues and ensure the reliability of the data. While the authors stated that they identified 13 EC clusters (Fig. 1B), expressing EC markers (Fig. 1C) but no other renal cell markers, it would be beneficial to present these data within a violin plot of normalized scRNA expression profiles of EC markers according to EC clusters and replace the current figure 1C

4-Improve the quality of figures by including scale bars. Some supplementary figures (S1, S2, S6) should be included in one figure (Validation) within the main text (Use only the marge image indicating the structures and the cells). Use marker to denote cells, and remark the structures. Please use magnification to highlight details. Include a summary illustration of nephron and vascular subtypes.

5-The section of results is full of references. This section should describe the obtained results rather than referencing the results of other studies. Use the discussion section to support your results with the results of others.

6-Include a paragraph discussing the novelty of your results and future trends, while avoiding speculation.

Author Response

Dear Reviewer,

We appreciate your constructive comments. Please find below our response. Thank you.

The authors studied temporal and spatial transcriptomics in various vascular compartments in a murine model of diabetic kidney disease. The paper is very interesting, but my initial enthusiasm was tempered by some methodological concerns and the presentation of results.

1-The authors exclusively used female diabetic BTBR ob/ob and the authors compared their results with a previous study published by David Barry (Nature communications 2019) that was conducted in 4-month male mice. The authors should justify their choice of using only females and consider the hormonal cycle of female mice could potentially influence the results.

We agree with the reviewer that gender considerations are important when conducting experiments. In general, studies with BTBRob/ob mice have used female mice for practical reasons as male mice in the in-house breeding are terminated at a very young age due to accelerated disease progression and development of unpredictable formation of urinary tract plugs that are fatal in some cases. We have treated female BTBRob/ob mice with several stand-of-care medicines, which showed translatable results. To map our cell populations from the single cell RNA-seq data to different renal endothelial cell types we utilized previously published markers, among them from Barry et al. if anything, our study shows that none of these markers are expressed differently based on gender. In addition, we did not intend to compare gene transcriptional changes quantitatively between our study and Barry’s study.

Additionally, the author mentioned that used another mice strain (C57Bl6/J) for validation purposes. However, the sex of mice used was not specified. It is important to justify the use of different strains for a validation experiment. Furthermore, the results of these experiments are not presented. The authors also used reporter mice with Cldn5GFP (Tg(Cldn5-GFP)Cbet/U) or Cdh5-TdTomato.  Was this control decided upon at the beginning of the study or was it included later?

It is important to note that the validation experiments were to validate specific endothelial cell populations within the kidney with IHC or RNA-ISH and not as a control for anything else (now Suppl. Fig. S1). The reporter mice were used as a tool in these validations, and that was decided when we had recognized markers that were needed and what mice we had available. Both female and male mice were used in these experiments, this has been added to Material and Methods. A conclusion from our studies is that there is no gender difference nor strain difference in the markers used to identify EC populations.

2-The Number of mice used is not clearly provided, and there is no evaluation of the kidney lesions of diabetic or measurement of creatinine and albuminuria. This data should be included in the first figure.

We have moved the number of mice used for scRNA-Seq in section 4.5 to section 4.1 in Materials and Methods to make it clearer. Endothelial cell populations in whole kidneys of these mice were quantified by FACS, which showed progressive vascular rarefaction and are now included in Fig 1. The mice used for scRNA-Seq were randomly selected from a breeding cohort. Several parameters including blood glucose, UACR, glomerular area are quantified in Suppl. Fig 5. Capillary rarefaction was also confirmed by Pecam1, Endomucin, and Podocalyxin immunofluorescence staining.

3-Regarding the scRNA-seq, the authors employed a pooling strategy that which may introduce challenges, such as potential cross-contamination between samples during library preparation or sequencing errors that could affect downstream analysis. Please, specify in the methods the quality control measures implemented to address these issues and ensure the reliability of the data. While the authors stated that they identified 13 EC clusters (Fig. 1B), expressing EC markers (Fig. 1C) but no other renal cell markers, it would be beneficial to present these data within a violin plot of normalized scRNA expression profiles of EC markers according to EC clusters and replace the current figure 1C.

We apologize for this misunderstanding; it should be noted that there was no pooling of actual samples. In this study, individual single cells were sorted into individual wells in 384-well plate format, followed by reverse transcription, PCR amplification and beads purification. Subsequently, the libraries were also prepared in 384-well plate format. Hence, all pooling was done at the analysis step on already sequenced samples. We did see some contaminating cells in our data set, and as stated in Material and Methods section 4.5. These contaminating cells were removed from analyses as they expressed high levels of epithelial cell markers including Lrp2, pax8, Slc12a1, Slc12a3. As per standard, cells with less than 50,000 read counts were removed (M&M, section 4.5.).

A violin plot of Figure 1E was generated to replace the UMAPs

4-Improve the quality of figures by including scale bars. Some supplementary figures (S1, S2, S6) should be included in one figure (Validation) within the main text (Use only the merge image indicating the structures and the cells). Use marker to denote cells and remark the structures. Please use magnification to highlight details. Include a summary illustration of nephron and vascular subtypes.

We apologize and have added scale bars to all merged and magnified images. As the reviewer suggested we have included a summary illustration of the vascular subtypes of the nephron in Fig. 1C. We have also included one of the validation images with the novel finding of Calca in efferent arteriole in Figure 1G to better entice the reader to look at additional validation data in Supp. Fig 1. We think Suppl. Fig 1 and 2 contain too much data to be include in the main text in a clear way, but they have been combined to Suppl. Fig. S1 and contain markers of different EC subtypes. On the other hand, Suppl. Fig S5 (former S6) contains quantification of EC density based on pan-EC marker immunostaining, which we belive it easier to follow by demonstrated in a separate figure.

5-The section of results is full of references. This section should describe the obtained results rather than referencing the results of other studies. Use the discussion section to support your results with the results of others.

The adding of references in Results is to help the reader find information about specific genes mentioned. We acknowledge that this is not commonly done (although some IJMS articles have it). However, we believe that it is necessary for the current manuscript that is rather complex.

6-Include a paragraph discussing the novelty of your results and future trends, while avoiding speculation.

This is a very good point and we have updated our Conclusion section in the Discussion to better highlight the novelty of our study.

Your sincerely,

Pernille B. Laerkegaard Hansen, Ph.D.

Executive Director, Head of Bioscience Renal

Research and Early Development Cardiovascular, Renal and Metabolism,

BioPharmaceuticals R&D, AstraZeneca

Pepparedsleden 1, Gothenburg, 43183, Sweden

Round 2

Reviewer 1 Report

Comments and Suggestions for Authors

The submitted responses are satisfactory 

Author Response

We thank the reviewer for your time on effort to improve our manuscript. 

Reviewer 3 Report

Comments and Suggestions for Authors

Thank you for the addressing my queries. However, I have some suggestions and concerns that need further improvement.

1-Animal model and gender need further consideration in the text:

In the introduction section, the authors need to justify the diabetes model used and the gender challenge.

1a-I am concerned about whether some differences depend on the animal model used and blood pressure. Authors stated that “Endothelial cell types in human kidney were highly conserved in the mouse kidney (ref 6) and the gene expression profiles of GECs and PCECs in our dataset correlated well with recently described human GECs and PCECs [ref 25]”. In my opinion, this point needs caution. Ref 6 also stated a divergence in hundreds of genes among mice (only available data from C57BL/6 mice) and humans. For example, CLDN15 showed an upregulation in mice and the gen Ace was not included.  Additionally, be cautious since Ref 25 was related to cancer, and I do not consider it a correct reference for your experiment.

1b- Please, clearly indicate in the figures when C57BL/6 mice were used and when BTBR ob/ob in validation experiments. Furthermore, include a comparison between diabetic BTBRob/ob mice and C57BL/6 in the maniscript.

1c-All these limitations have to be included in the discussion sections.

2-In the Results section, it was stated, “For EC subtype annotation (Fig. 1FD), we compared cluster-specific gene patterns with published renal EC scRNAseq data and protein and mRNA localization in situ [3-5]”. I suggest including in the introduction section a table with main markers of subtypes of endothelial cells, the experimental model used (including gender and age) or whether the results were from patients (including references to public databases) and if the results were validated.

3-Please provide justification for the number of mice used and the power analysis conducted in the methods section.

4-Quality of the images. Ensure that all images have scale bars and consider using arrows to indicate positive cells to help readers.

5-In my opinion, the BTBR ob/ob phenotype (Figure S5: validation of DKD with blood glucose, urinary albumin creatinine ratio) should be included in the main text. The authors should also include data from Blood pressure (if BP was not measured, this limitation has to be discussed).

I understand that some journals combine results and discussion, but I would prefer that you solely focus on describing your own results obtained in your study. Then, use the discussion section to provide support and context for these results.

Author Response

We thank the reviewers for their time on effort to improve our manuscript. We have replied to the comments below.

Thank you for the addressing my queries. However, I have some suggestions and concerns that need further improvement.

1-Animal model and gender need further consideration in the text:

In the introduction section, the authors need to justify the diabetes model used and the gender challenge.

We have now clarified the model in the last paragraph of Introduction with the following: “The leptin deficient BTBRob/ob mice features hyperglycemia, albuminuria, obesity, and hyperlipidemia, but not hypertension (PMID: 33911188, PMID: 24944269, PMID:20634301)”

Gender challenge was included as limitations in the Discussion with the following sentence: “First, only female BTBR mice were included in the study due to rapid disease progression and severe phenotypes in male BTBRob/ob mice causing loss of animals in breeding. Potentially gender-specific transcriptomic changes were not studied.”

1a-I am concerned about whether some differences depend on the animal model used and blood pressure. Authors stated that “Endothelial cell types in human kidney were highly conserved in the mouse kidney (ref 6) and the gene expression profiles of GECs and PCECs in our dataset correlated well with recently described human GECs and PCECs [ref 25]”. In my opinion, this point needs caution. Ref 6 also stated a divergence in hundreds of genes among mice (only available data from C57BL/6 mice) and humans. For example, CLDN15 showed an upregulation in mice and the gen Ace was not included.  Additionally, be cautious since Ref 25 was related to cancer, and I do not consider it a correct reference for your experiment.

According to previous reports BTBRob/ob mice do not have increased blood pressure. We have added the following sentence to the Discussion “The absence of hypertension in this model allows identification of mediators of diabetic kidney disease independent of the confounding effects of coexisting hypertension as may occur in other models”.

We understand the concern regarding the comparison between mouse and human data. We have removed “highly” from the sentence above and added “in spite of certain species-specific genes” in the sentence.

In terms of human gene expression profile, we have only utilized the control cells in Ref 25 (now ref 27), which were from areas visually absent from tumor tissue and considered to be normal.

Regarding gene Ace, we have stated in Results 2.2 that “We identified Ace as a novel marker for AA, as confirmed by RNA-ISH (Suppl. Fig. 1B)”.  However, Ace expression was not significantly regulated in AA in disease.

1b- Please, clearly indicate in the figures when C57BL/6 mice were used and when BTBR ob/ob in validation experiments. Furthermore, include a comparison between diabetic BTBRob/ob mice and C57BL/6 in the manuscript.

All figure legends now include clear indications of what mouse strain was used. In our study C57BL6/J mice were only used for in situ hybridisation validation of certain markers, but not for scRNAseq. While renal endothelial scRNAseq analyses in C57BL6/J mice have been performed in previous studies (Ref 3 & 4). Based on our validation data, the selected EC subtype markers were conserved between the two mouse strains. On the other hand, a comprehensive comparison of gene expression profiles between the two mouse strains is beyond the scope of the current study, which could be of interest for a meta-analysis study.

1c-All these limitations have to be included in the discussion sections.

Each point has been addressed above.

2-In the Results section, it was stated, “For EC subtype annotation (Fig. 1FD), we compared cluster-specific gene patterns with published renal EC scRNAseq data and protein and mRNA localization in situ [3-5]”. I suggest including in the introduction section a table with main markers of subtypes of endothelial cells, the experimental model used (including gender and age) or whether the results were from patients (including references to public databases) and if the results were validated.

The main focus of our study is transcriptomic changes during progression of DKD. That being said, we do need to distinguish different EC subtypes in order to assess subtype-specific changes. Identification of EC subtypes was studied in physiological condition. A small number of selected markers were clearly demonstrated Suppl. Fig 1. We are of the opinion that a comprehensive analysis of EC subtype markers across mouse strains and species is of more interest for a meta-analysis study.

3-Please provide justification for the number of mice used and the power analysis conducted in the methods section.

SmartSeq2 technique allowed us to perform deep sequencing to explore lower abundant genes and full transcript sequencing to explore splicing variants. However, it is not as high throughput as droplet-based scRNAseq techniques. In this study, single cells of the same subtype in the same group were pooled for DEG analysis, which was stated in Result section 2.4. Including 3 mice per group is to ensure robust transcriptomics data within the group. No statistical method was used to predetermine sample size. We have now added more information to 4.13 Statistical analysis “To identify DEGs, the individual cells between groups were compared using the Wilcoxon Rank Sum test. Multiple test correction was performed using Bonferroni method and the corrected p value < 0.05 was set as significance.”.

4-Quality of the images. Ensure that all images have scale bars and consider using arrows to indicate positive cells to help readers.

We have now added scale bars to all images and arrows to several images to aid the reader.

5-In my opinion, the BTBR ob/ob phenotype (Figure S5: validation of DKD with blood glucose, urinary albumin creatinine ratio) should be included in the main text. The authors should also include data from Blood pressure (if BP was not measured, this limitation has to be discussed).

Blood glucose and urinary albumin/creatinine ratio (former Suppl. Fig. S5A, B) are now included in Figure 1 (C, D). The figure legend and Results have been updated accordingly.

Blood pressure was not measured in the current study. However, the in house breeding cohort has been phenotyped previously without showing hypertension, which is consistent with published information in this model. As mentioned above, the blood pressure has been added to the introduction and Discussion.

I understand that some journals combine results and discussion, but I would prefer that you solely focus on describing your own results obtained in your study. Then, use the discussion section to provide support and context for these results.

We respect the reviewer’s different opinion, but we still think that the references in Results help the reader find information about specific genes mentioned. A typical scRNAseq study usually involves reporting a large number of genes. In Results we have discussed multiple specific markers and DEGs, while in Discussion, we prefer to focus on the most significant biological pathways and the overall message. We are concerned that a detailed discussion of many individual genes in the Discussion section would be distracting to readers.

Round 3

Reviewer 3 Report

Comments and Suggestions for Authors

I agree with the answers and with the improvements to the text.